# Empirical robustification of pre-trained classifiers

**Mohammad Sadegh Norouzzadeh** [1]   **Wan-Yi Lin** [1]   **Leonid Boytsov** [1]   **Leslie Rice** [2]   **Huan Zhang** [2]
**Filipe Condessa** [1]   **J. Zico Kolter** [1,2]

## Abstract

Most pre-trained classifiers, though they may work extremely well on the domain they were trained upon, are not trained in a robust fashion, and therefore are susceptible to adversarial attacks. A recent technique, denoised-smoothing, demonstrated that it was possible to create certifiably robust classifiers from a pre-trained classifier (without any retraining) by pre-pending a denoising network and wrapping the entire pipeline within randomized smoothing. However, this is a costly procedure, which requires multiple queries due to the randomized smoothing element, and which ultimately is very dependent on the quality of the denoiser. In this paper, we demonstrate that a more conventional adversarial training approach also works when applied to this robustification process. Specifically, we show that by training an image-to-image translation model, prepended to a pre-trained classifier, with losses that optimize for both the fidelity of the image reconstruction and the adversarial performance of the end-to-end system, we can robustify pre-trained classifiers to a higher empirical degree of accuracy than denoised smoothing, while being more efficient at inference time. Furthermore, these robustifers are also transferable to some degree across multiple classifiers and even some architectures, illustrating that in some real sense they are removing the "adversarial manifold" from the input data, a task that has traditionally been very challenging for "conventional" preprocessing methods.

This work is sponsored by DARPA grant HR11002020006 [1]Bosch Center for Artificial Intelligence, Pittsburgh, USA [2]Carnegie-Mellon University, Pittsburgh, USA. Correspondence to: Mohammad Norouzzadeh <arash.norouzzadeh@us.bosch.com>.

*Accepted by the ICML 2021 workshop on A Blessing in Disguise: The Prospects and Perils of Adversarial Machine Learning.* Copyright 2021 by the author(s).

## 1. Introduction

Deep classifiers are well-known to be sensitive to adversarial attacks, small perturbations of input images that can drastically change the output of the network. Although there exist methods for improving the robustness of these classifiers to such attacks (e.g., adversarial training (Madry et al., 2017) or certifiably robust training (Wong & Kolter, 2018; Salman et al., 2019)), most pre-trained models do not adopt such training strategies, as they often come at the cost of substantial drop in clean accuracy. As a field, however, deep learning is also moving toward the adoption of larger and larger (and more and more costly to train) pre-trained models, raising the question of whether it might be possible to "robustify" these existing models without having to retrain them from scratch using a robust training approach.

Unfortunately, attempting to naively robustify a pre-trained model is a challenging task. Most previous methods that attempted to use e.g., simple image preprocessing techniques were broken by simple adaptive attacks that attempted to attack that end-to-end system of both the preprocessor and the classifier. A recent advance in this area, however, came from the development of Denoised smoothing (Salman et al., 2020). Denoised smoothing exploits the fact that randomized smoothing (Cohen et al., 2019) can transform classifiers that are robust to *random* perturbations to ones that are robust to ($\ell_2$ bounded) adversarial perturbations; thus, denoised smoothing prepends a denoiser to the pre-trained classifier, and applies randomized smoothing to the combination, leading to a (certifiably) robust meta-classifier. Unfortunately, denoised smoothing has substantial drawbacks as well: because it employs randomized smoothing, it requires *many* queries to the underlying classifier to determine the prediction of the robustified classifier. It also is not easily extensible to alternative perturbation models such as $\ell_\infty$ perturbations. This is also why, for instance, strong adversarial training is often used in practice over randomized smoothing or other certifiably robust models: to achieve a higher degree of empirical robustness at the cost of guarantees.

**Our contribution**   In this paper, we propose an "empirical robustness" analogue of these methods for robustifying pre-trained models, demonstrating an efficient yet powerful

method to make a pre-trained classifier robust to adversarial attacks without re-optimizing any of its parameters. Specifically, we propose to prepend an image-to-image translation model (called the robustifier) in front of the pretrained model. However, unlike past attempts at creating robust models by preprocessing images (which typically looked only at attacks on the original model), we train the robustifier so as to optimize the adversarial performance of the entire end-to-end pipeline (i.e. the combination of the robustifier and the pretrained model); we do this, however, by *only* modifying the parameters of the robustifier itself, and leaving the parameters of the pre-trained model frozen. Also key to our approach is the inclusion of several different loss terms, which penalize 1) the adversarial loss of the end-to-end system (to ensure the end-to-end system is robust); 2) the difference between input and outputs of the robustifier (to ensure that the robustifier doesn't destroy the image content); and 3) the difference in activations between original and adversarial images in the pretrained classifier (this enforces that the post-robustified image should "look like" the non-adversarial image as much as possible to the classifier). We demonstrate that:

- Our defense not only eliminates the need for multiple queries per sample but also provides better results than denoised smoothing. The approach can also in theory be applied to any setting where adversarial training can be used, rather than applying only to classification settings.

- It can be transferred among different models and even different architectures. The transferability results indicate that a preprocessor model can—-to some extent— learn to separate the "adversarial manifold" from the input data, a key challenging task for "conventional" preprocessing methods.

## 2. Related work

Many earlier front-end defenses relied on heuristic preprocessing of input images, which shatter gradients and make a straightforward application of a vanilla gradient attacks impossible. For example, (Guo et al., 2017) employed multiple image transforms including cropping, tilting, bit-depth reduction. However, such simple image transformations were shown to be ineffective under the Backward Pass Differentiable Approximation (BPDA) attack (Athalye et al., 2018).

More sophisticated front-end defenses attempt to detect and/or remove adversarial noise using neural networks. An attempt to detect and remove adversarial noise using sophisticated detector and reformer networks was proposed by (Meng & Chen, 2017), but this defense was easily broken under adaptive attacks (Carlini & Wagner, 2017b). (Saman-

gouei et al., 2018) harnessed generative neural networks in an attempt to remove adversarial noise, but it was also shown to have lower effectiveness under adaptive attacks (Athalye et al., 2018). Similarly, (Mustafa et al., 2019) proposed a DNN based image super-resolution method. However, (Choi et al., 2019) showed that image super-resolution models were also vulnerable to adversarial attacks. Additionally, (Carlini & Wagner, 2017a) showed that most detectors for adversarial examples are ineffective. Although these works use an approach similar to our defense (transforming input images before feeding them into a classifier), they are mostly based on heuristics and cannot truly remove the "adversarial manifold" generated by strong and adaptive attacks.

Recently, (Salman et al., 2020) proposed an effective method to make a pre-trained classifier robust by equipping it with a custom-trained Gaussian denoiser. To classify an input image, first the input image is perturbed with Gaussian noise multiple times and fed to the model. Then, randomized smoothing (Cohen et al., 2019) is employed to determine the class of the input image. Their method certifies the robustness of a model without changing model parameters under white-box or black-box attacks. The guarantee of the degree of robustness for any classifier leveraging denoised smoothing comes at the cost of having to run the model multiple times (10 to 100) for each input, which greatly increases inference time.

## 3. Robustifier models for pre-trained networks

In this section, we discuss the proposed pre-pended *robustifier* to make a pre-trained machine learning model robust against adversarial attack (Fig. S.1). Different from (Salman et al., 2020), the proposed method requires only one forward pass and can be directly applied to tasks beyond classification. We begin by reviewing the standard adversarial training setting, then describe the extension to our robustifier setting, as well as the additional loss terms we found to be crucially important for good performance of the method. We finally discuss how to train the robustifier to improve transferability of the robustifier between multiple different variants of an architecture and even between different architectures.

### 3.1. Standard and adversarially robust training

Given a classifier $h_\theta : \mathcal{X} \to \mathcal{Y}$ mapping from an input space $\mathcal{X}$ to output space $\mathcal{Y}$, parameterized by parameters $\theta$, the typical goal of training is to minimize some expected loss function

$$\min_\theta \ \mathbf{E}_{x,y \sim \mathcal{D}} \left[ \ell(h_\theta(x), y) \right] \tag{1}$$

where $\mathcal{D}$ denotes a distribution over $x, y$ pairs, and $\ell : \mathcal{Y} \times \mathcal{Y}$ denotes some loss function. *Adversarially robust* training

(Madry et al., 2017), on the other hand, seeks to minimize the *worst case* loss when inputs are allowed to be perturbed within some pre-specified threat model $\Delta$ (often taken to be e.g., an $\ell_p$ norm ball, though other threat models are certainly possible as well). This can be written formally as the optimization problem

$$\min_{\theta} \ \mathbf{E}_{x,y \sim \mathcal{D}} \left[ \max_{\delta \in \Delta} \ell(h_\theta(x + \delta), y) \right]. \tag{2}$$

A standard approach to optimizing this robust objective is via adversarial training, which for each sample in a dataset drawn from $\mathcal{D}$ iteratively seeks to solve the maximization problem over $\delta$ (via a gradient-based approach like projected gradient descent), and then takes a gradient step in $\theta$ at these worst-case perturbations.

### 3.2. Training prepended robustifiers

Given a fixed pre-trained classifier $h : \mathcal{X} \rightarrow \mathcal{Y}$ (we no longer write $h$ as dependent on parameters $\theta$, as we will treat $h$ as fixed throughout), we propose to simply prepend an image to image translation system we refer to as the *robustifier*. Formally, a robustifier is a network $r_\theta : \mathcal{X} \rightarrow \mathcal{X}$, parameterized by parameters $\theta$ such that our final prediction is given by the composition of the classifier and robustifier $h(r_\theta(x))$. Generally speaking, any image-to-image architecture could be used as the robustifier, such as Variational Autoencoders (VAEs), image denoisers, or semantic segmentation networks (we will ultimately use a U-Net style architecture for this task).

To train the robustifier, we could apply adversarial training via projected gradient descent (PGD) to composed system of the robustifier and the classifier. That is, we could perform adversarial training on the objective

$$\min_{\theta} \ \mathbf{E}_{x,y \sim \mathcal{D}} \left[ \max_{\delta \in \Delta} \ell(h(r_\theta(x + \delta)), y) \right]. \tag{3}$$

where $\ell$ is the cross entropy loss, and $\Delta$ is the perturbation model. The only difference between this and adversarial training is that the the weights of the classifier itself are fixed, and we are only using the parameters of the robustifier.

However, if we train the robustifier with this objective, without any constraints on the output of the image-to-image network, the output images can diverge too far from the original images (Nguyen et al., 2015). Effectively, because the robustifier can learn to output *any* image into the classifier, the robustifier can effectively identify any low-dimenional manifold of the desired *output* itself, and simply output images in this reduced manifold: this raises the possibility that the robustifier could simply "duplicate" the effort of a typical robust classifier, rather than actual "filtering" away any aspects of the adversarial perturbation in a generic manner.

To avoid this phenomenon, we additionally need to ensure some measure of fidelity to the original image. We can do this by additionally incorporating some form of recontruction loss in our training objective, such as the Mean Squared Error (MSE) loss. Let $\tilde{x}$ denote the adversarial perturbation of $x$ according to the maximization in 3. Then we can write this MSE loss as

$$\ell^{\mathrm{mse}}(x, \tilde{x}) = \|x - r_\theta(\tilde{x})\|_2^2. \tag{4}$$

To further improve in this regard, we found in practice it was additionally beneficial to penalize an MSE loss between intermediate layer activations of the pre-trained classifier applied to the original image, and these intermediate layer activations when applied to the output of the robustifier (applied to the adversarial example). That is, we want the output of the robustifier, when applied to an adversarial image, to produce an image that is "nearby" the original image, both in terms of the MSE on the original image *and* in terms of the activations produced by the pretrained classifier. Letting $h_a$ denote the activations of classifier $h$ at selected intermediate layers $L$, this corresponds to the loss

$$\ell^{\mathrm{act}}(x, \tilde{x}) = ||h_a(x) - h_a(r_\theta(\tilde{x}))||_2^2. \tag{5}$$

Altogether, our training objective is the following:

$$\alpha \ell(h(r_\theta(\tilde{x})), y) + \beta \ell^{\mathrm{mse}}(x, \tilde{x}) + \gamma \ell^{\mathrm{act}}(x, \tilde{x}) \tag{6}$$

where the coefficients $\alpha, \beta$, and $\gamma$ adjust the scale and relative importance of the loss terms.[1] It is important to emphasize that while we include all three loss terms in training the robustifier, the adversarial attack itself does not directly try to maximize this combined loss, but just maximizes the original robust loss: this is crucial in practice, as the "strongest" adversary (and our eventual evaluation metric) is concerned solely with maximizing the adversarial loss; the other two components are effectively regularization terms, that prevent the robustifier model from overfitting to the min-max objective. For more details on implementation of training procedure refer to section S.1.

## 4. Experiments

In this section, we present experiment setup and then discuss the obtained results when training the robustifier with a single pre-trained classifier. Sec. S.1 of the supplementary materials contains details about our implementation, dataset, and architecture of models. We discuss training the robustifier with an ensemble of classifiers and transferability of the robustifier in supplementary materials Sec. S.4.

---

[1]While of course in practice we could remove of these three coefficients and have the other two be coefficients relative to the third, in practice we found it slightly more convenient to explicitly normalize each of the three components separately.

## 4.1. Robustifier trained with one classifier

As discussed in Section 3.2, we follow two main objectives during training: 1- adversarial robustness loss (cross-entropy loss), and 2 - fidelity to the original image (MSE loss and the intermediate layers loss). In extreme cases, if we abandon the adversarial robustness loss ($\alpha = 0$ in Eq.6), we get lower robust accuracy/higher clean accuracy, and the output of the robustifier will be similar to the original image. On the other hand, if we abandon the fidelity objective ($\beta = 0, \gamma = 0$ in Eq. 6), we get higher robust accuracy/lower clean accuracy, and the outputs of the robustifier could be heavily distorted. We run different experiments to find the best set of loss coefficients to make a pre-trained classifier robust. Section S.3 of supplementary materials describes the steps we perform to find an appropriate set of coefficients. Fig. 1 depicts the achieved robust and clean accuracy under 100 steps PGD $\ell_\infty$ and $\ell_2$ attacks for several good performing robustifiers using the CIFAR-10 dataset.

**Comparison against denoised smoothing** Denoised smoothing (Salman et al., 2020) is the most relevant work to ours as both methods try to make a pre-trained classifier immune to adversarial attacks although they reported certified accuracy (Cohen et al., 2019) in the paper. To compare our approach to denoised smoothing fairly, we conduct experiments on the empirical robustness of denoised smoothing for different $\sigma$ values and denoised objectives (MSE and stability), and then compare the best results of both methods. Empirical results of denoised smoothing for $\ell_\infty$ and $\ell_2$ attacks with different denoiser objectives are depicted in supplementary materials S.2. As the results reveal, the best setting for denoised smoothing achieves 72% clean, 32% robust accuracy under $\ell_\infty$ attack, and 53% robust accuracy under $\ell_2$ attack, shown as black dash line in Fig. 1(b) and Fig. 1(c). These results are inferior to the results we obtained as in Fig. 1. This shows that our approach can offer a higher empirical robust accuracy than denoised smoothing while avoiding the high computational cost of multiple forward pass during inference time.

## 5. Conclusion, limitation, and broader impact

In this paper, we demonstrate an approach that uses adversarial training to robustify existing pre-trained classifiers, without any retraining or fine-tuning of the classifier itself. We accomplish this by prepending a custom-trained image-to-image model to these pre-trained classifiers and then adversarially training the prepended model while preserving a measure of fidelity to the input image. We evaluated U-Net-based robustifier models at different capacities and different coefficients of losses to robustifier models that strike a good balance between robust and clean accuracy levels. Moreover, we showed our defense strategy is transferable among different models and even different architectures, revealing the power of our suggested method and shedding light on common vulnerability patterns of different deep learning models.

Despite theses benefits, the approach we present here also has some significant limitations. Most notably, the robust accuracy we achieve still does lag substantially behind the state of the art in adversarially robust training. For instance, state of the art methods in adversarial training (albeit with a larger model and using additional data augmentation techniques), achieve great than 65% robust accuracy on CIFAR10 (Gowal et al., 2020). Although we imagine that some of these methods could be used to improve robustifier performance, there will likely still be a substantial gap between our robustifier method and models trained from scratch via adversarial training. To some degree, we believe this to be unavoidable: existing pre-trained classifiers *are* susceptible to adversarial attacks, and any attempt to robustify them still needs to cope with the reality that a fixed part of the model cannot be changed to be inherently more robust. But the results we present here, especially the transferability results, nonetheless *do* suggest that some real adversarial component is being removed from the data, and further study of this "manifold" may prove fruitful in future work.

Our hope is that the work here can largely serve to benefit existing models by making "more robust" versions, without having to sacrifice the accuracy of the underlying classifier (e.g., when the robustifier is removed). Thus one could e.g., add a number of "plug in components" that allowed a user to select the desired level of robustness needed for a given classifier, or even adapt in real-time. In addition, our transferability results indicate researchers do not even need to train a custom robustifier for their model and can use a pre-trained robustifier to make their model robust. As such, we hope that the work would have substantially more positive societal benefits than negative ones. Adversarial attacks can endanger many real-world applications of computer vision, including autonomous driving cars and other safety-critical uses, and this system can hopefully seek to improve their performance.

At the same time, it is also worth emphasizing the potential negative consequences of adopting a "robust" model, especially in the case that the robust accuracy is still not nearly on par with modern deep learning systems. In particular, adopting a "robustifier" model may provide some degree of a false sense of security, that the sensitivity of the model to adversarial attacks has been "solved", when in reality even if the best adversarially robust models still substantially underperform the best clean models, and worse are easy still attacked by perturbations outside their specified threat model. This is all to say that the task of truly robust machine

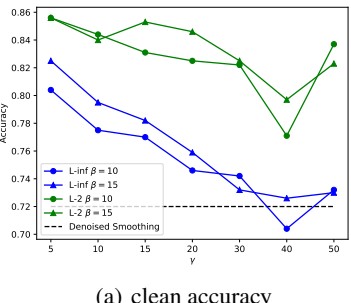

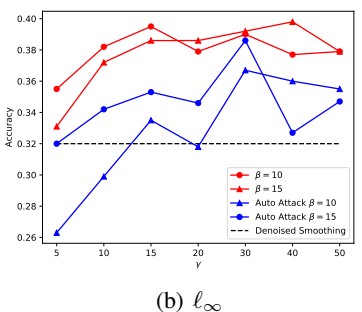

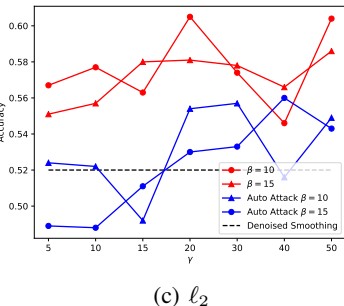

(a) clean accuracy

(b) $\ell_\infty$

(c) $\ell_2$

Figure 1: Clean accuracy (a), robust accuracy for $\ell_\infty$ norm (b), and robust accuracy for $\ell_2$ norm (c). We set $\alpha = 0.005$ for all the experiments. The dashed lines show the best accuracy for the denoised smoothing method (Salman et al., 2020). For the robust accuracy we report both 100 steps PGD attack accuracy and the results of Auto Attack method (Croce & Hein, 2020).

learning has a great deal of progress left to be made, and the ultimate societal impact of the field is still evolving.

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

# Supplementary materials

## S.1. Implementation

In order to minimize over the objective 6 in practice, when we sample a minibatch $(x, y)$ from the dataset, we first duplicate the minibatch, and then adversarially perturb one of the copies using PGD to get $\tilde{x}$. We then feed both $x$ and $\tilde{x}$ to the robustifier and respectively feed the outputs of the robustifier, $r_\theta(x)$ and $r_\theta(\tilde{x})$, to the pre-trained classifier $h$. We store $r_\theta(x)$ and $r_\theta(\tilde{x})$, the final output of the pre-trained classifier for the perturbed image, $h(r_\theta(\tilde{x}))$, as well as activations of selected intermediate layers during the forward path, $h_a(x)$ and $h_a(r_\theta(\tilde{x}))$. Finally we compute the losses according to the objective in 6 and back-propagate the error. Figure S.2 illustrates the training procedure.

**Dataset, threat model, and architecture** We utilize the CIFAR-10 dataset to evaluate the proposed methods. We investigated both $\ell_\infty$ and $\ell_2$ PGD attacks with the radius of 0.031 and 0.5, respectively. For training, we use 5-step PGD with the step size of 0.008, and for evaluation, we use 100 steps PGD (5 restarts, 20 steps each) with different step sizes (0.008, 0.004, 0.0015) and report the lowest obtained robust accuracy. We also report robust accuracy under Autoattack (Croce & Hein, 2020) for robustifiers trained with one pre-trained classifier. We employ a Momentum Stochastic Gradient Decent (SGD) optimizer utilizing one cycle learning rate policy (Smith, 2017) during training with the maximum lr =0.005, momentum=0.9, weight decay=0.0005, and 20% steps spent increasing the learning rate. We adversarially train the robustifiers on one RTX 2080 GPU for 100 epochs with early stopping; each epoch takes about 6 minutes.

For the robustifier, we chose to use an architecture similar to that of U-Net (Ronneberger et al., 2015). The U-Net architecture is advantageous because of its skip connections, which enable the model to reconstruct fine details of input image. Details on the U-Net model we used can be found in the next paragraph. For the pre-trained classifier, we use ResNet-18 architecture.

**U-Net architecture** U-Net is a fully convolutional neural network architecture designed for segmentation and image translation (Ronneberger et al., 2015). The skip connections in U-Net architecture enable it to remember fine details of the input image while extracting the high-level semantic of the image using its back-bone. In this paper, we employ a customized U-Net architecture which is depicted in Fig. S.4. The scale factor parameter specifies the number of convolutional kernels at each layer, affecting the number of total parameters in the model. We use the scale factor of 4, 8, and 16 with 4.3, 17.3, and 69.0 million parameters, respectively.

## S.2. Empirical results of denoised smoothing

Throughout this paper, we use the empirical results of denoised smoothing (Salman et al., 2020) as our baseline. We obtained empirical results of denoised smoothing (Salman et al., 2020) with fine-tuned DnCNN denoiser with MSE and stability objectives in Table S.1 and Table S.2. Here the attacks are 100-PGD (50 iterations, 2 restarts).

## S.3. Tradeoff between loss coefficients

As equation 6 indicates, our loss function consists of three components with their own coefficients $\alpha$, $\beta$, and $\gamma$. Finding a good equilibrium between these coefficients is crucial for achieving a robustifier model which removes the adversarial perturbation while keeping the image similar to the input image. Trying all the possible coefficients is computationally impossible. Therefore, to find the best set of coefficients, we conduct a grid search on a range of appropriate coefficient values. First, we specify an appropriate range of the coefficients based on a few sample runs and then perform a grid search on those ranges. Fig. S.3 depicts the results of the grid search. For all the experiments in Fig. S.3 we utilize U-Net with scale factor 4 because it has fewer parameters, and hence it is faster to train and evaluate. After obtaining the results of grid-search, we pick the two most promising rows of figure S.3 ($\alpha = 0.005$, $\beta = 10, 15$ and $\gamma = 5, 10, 15, 20, 30, 40, 50$) and redo the experiments with the scale factor of 16 ( Fig. 1). Fig. 1 confirms our intuition that if we fix weight for classifier loss, $\alpha$, and increase the weights of fidelity loss, $\beta$ and $\gamma$, the robust accuracy increases as clean accuracy decreases. We also briefly try the scale factor of 8 for four of our best coefficient sets ($\alpha = 0.005$, $\beta = 10$ and $\gamma = 20, 30, 40, 50$) and report the results in Figure S.5.

To further verify the robustness of our robustifier defense, we also report robust accuracy of our models under Autoattack (Croce & Hein, 2020) as the two blue lines in Fig. 1(b) and (c). Our best setting, $\alpha = 0.005$, $\beta = 15.0$, $\gamma = 30.0$ can achieve 39.0% PGD, 38.4% Autoattack robust accuracy and 74.2% clean accuracy for $\ell_\infty$ attack; and 57.3% PGD, 55.6% Autoattack robust accuracy, and 82.5% clean accuracy for $\ell_2$ attack.

## S.4. Transferability of the robustifier

Adversarial examples generated for one model can also be effective on another model trained with different initialization, or even models with different architectures (Liu et al., 2016; Papernot et al., 2017). This is commonly referred to as *transferability* of adversarial examples. (Su et al., 2018) showed that adversarial examples transfer well between commonly used image classifiers architectures, such as ResNet (He et al., 2016), DenseNet, VGG and Inception.

Robustified Model

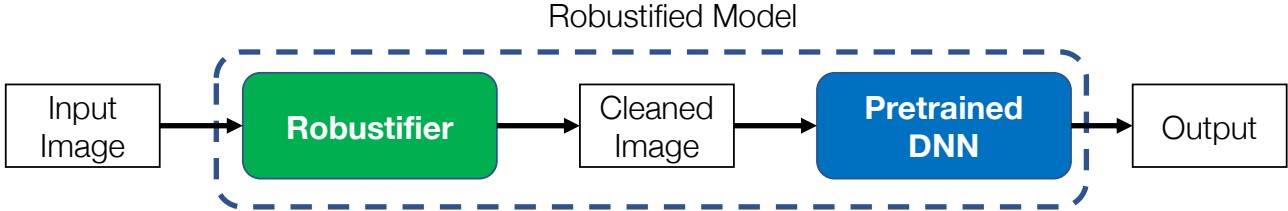

Figure S.1: Our robustifier models translates a given input image to another image which is optimized to generate the correct label when fed to a pretrained classifier. The robustifier is trained using adversarial training (Madry et al., 2017).

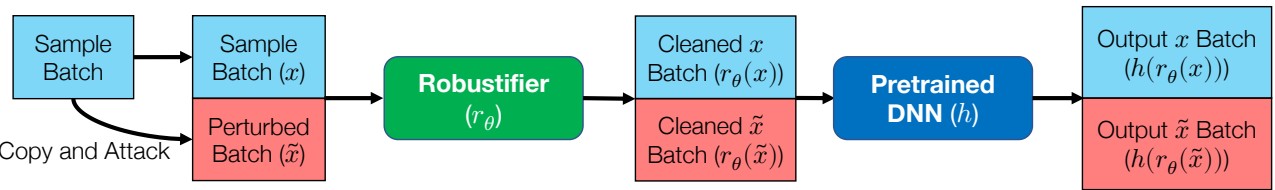

Figure S.2: Both clean and perturbed versions of each batch are fed to the robustifier model and consequently the pre-trained classifier. The loss function consists of the cross-entropy loss, MSE between the input images and output of robustifier for perturbed images, and MSE loss between activation values at different selected layers of the pre-trained classifier for the outputs of the robustifier. Loss value is calculated using equation 6.

The transferability of adversarial examples indicates that the adversarial manifold of different model architectures are closely related, thus if we can remove adversarial noise on one model using robustifier, it is likely to be effective on models trained with different initialization, as well as other similar model architectures.

To improve transferability of robustifier, we propose to train the robustifier with an ensemble of $M$ classifier (e.g., 20 ResNet-18 models trained using different initializations, or a few classifiers with different model architectures). To avoid the cost of forward and backward propagation on $M$ base classifiers, we propose to just randomly sample one base classifier each time rather than adding the prediction of all $M$ classifiers together. This allows us to train the robustifier using an ensemble of base classifiers, while keeping the training cost similar to the case of training with one base model. We evaluate the performance and transferability of robustifier when trained with an ensemble classifier in Section 4.

### S.4.1. Transferability Experiments

In this section, we investigate the transferability of our proposed robustifier, by training the robustifier on an ensemble of 2, 4, 20 ResNet-18 models on CIFAR-10 dataset and then evaluate it on a few settings: (1) a ResNet-18 model that is included in the ensemble (denoted as "ResNet-18 inc." in Table S.3 and Table S.4), (2) a ResNet-18 model that is not included in the ensemble (denoted as "ResNet-18 exc." in Table S.3 and Table S.4), and (3) different model

architectures such as ResNet-34, ResNet-50, DenseNet-121, and VGG-16. In each setting, we evaluate the robustness of the robustifier and classifier end-to-end PGD attack as described in Section 4. The aim of this experiments is to show the generalization of robustifier and its effectiveness on a wide range of popular model architectures.

We show the transferability results in Table S.3 for $\ell_\infty$ attack and Table S.4 for $\ell_2$ attack. The robustifier trained on multiple ResNet-18s (row 4-11 of Table S.3 and Table S.4) also transfer to other ResNet-18 models, and the PGD accuracy almost stays the same, which demonstrates that the robustifier does generalize to unseen model parameters. Furthermore, robustifiers trained on ensemble of Resnet-18, even with just two of them, also transfers reasonably well to other classifier architectures such as DenseNet and VGG. As the number of classifier models used for training a robustifier increases, the transferability of the robustifier increases but not significantly. However when the robustifier was trained on one ResNet-18 (row 2-3 of Table S.3 and Table S.4), it does not transfer to other ResNet-18s as well as other classifier architectures.

**Impact of activations of intermediate layers used for fidelity loss** In addition to the loss coefficient $\alpha, \beta, \gamma$, the set of intermediate layers used for the fidelity objective also influences the tradeoff between robust accuracy and clean accuracy as well as visual quality of the robustifier output. If we use the activations of last few layers of the robustifiers for Eq. 5, then the visual quality of the robustifier output is poorer with lower clean accuracy and higher robust accu-

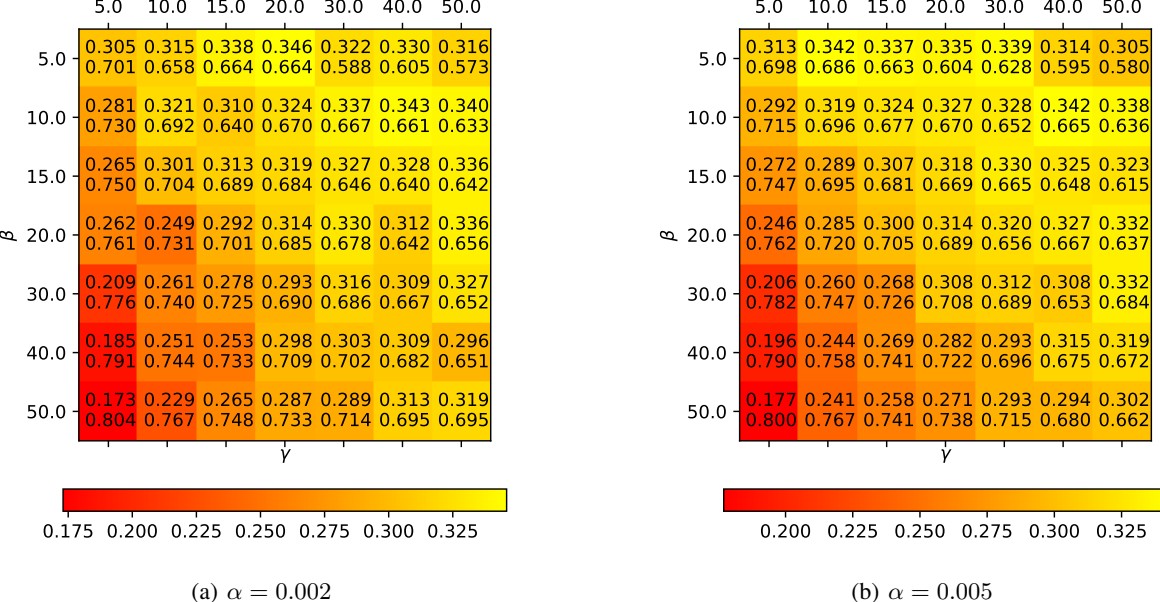

(a) $\alpha = 0.002$                                          (b) $\alpha = 0.005$

Figure S.3: Results of grid search to find best values for the $\alpha$, $\beta$ and $\gamma$ coefficients for U-net robustifier with scale factor = 4. The top number in each cell is the robust accuracy and the bottom number is the clean accuracy. The colors indicate the robust accuracy level. Higher values of $\beta$ tend to increase clean accuracy and decrease robust accuracy while higher value of $\gamma$ encourage higher robust accuracy and lower clean accuracy.

racy. Visual quality of robustifier output with corresponding accuracy can be found in Fig. S.6: Fig. S.6(b) uses activations at the end of each of the four blocks of the ResNet-18 architecture to compute Eq. 6, while Fig. S.6(c) additionally include the activations of the final fully connected layer. It is clear that the visual quality of Fig. S.6(c) is almost the same as the input (Fig. S.6(a)) and is much better than Fig. S.6(b); the robustifier that produces Fig. S.6(c) has lower robust accuracy (31%) than the robustifier that produces Fig. S.6(b) (39%) under $\ell_\infty$ attack but higher clean accuracy (87% vs 78%).

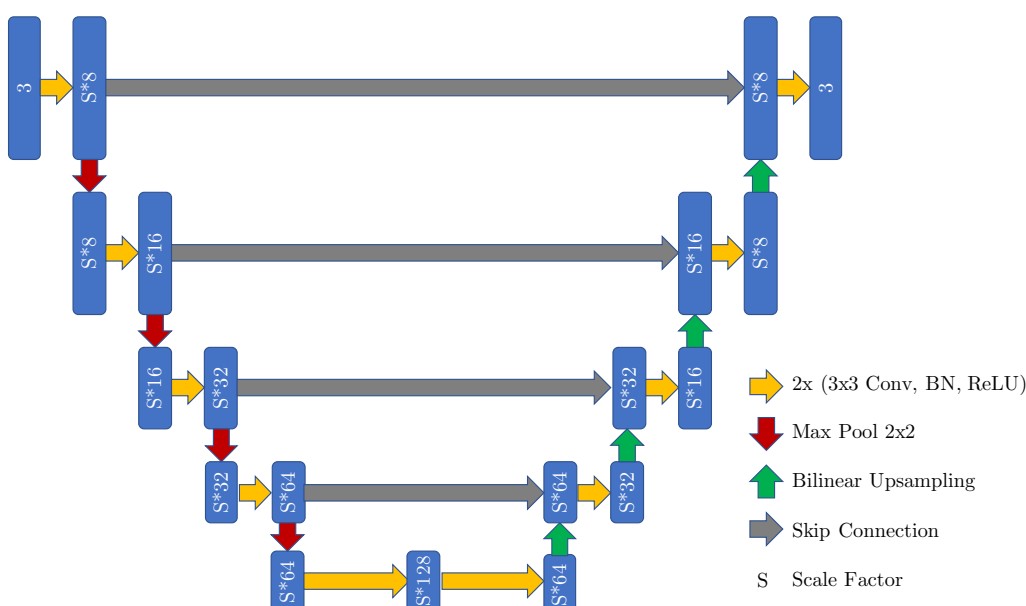

Figure S.4: We utilized a tailored U-Net architecture in our experiments. Yellow arrows depict two consecutive 3x3 convolutional layers, each followed by a batch-norm and a ReLU activation layers. Red arrows are 2D 2x2 max-pooling layers, and green arrows are bilinear upsampling layers. S indicates the scale factor parameter, and gray arrows illustrate skip connections.

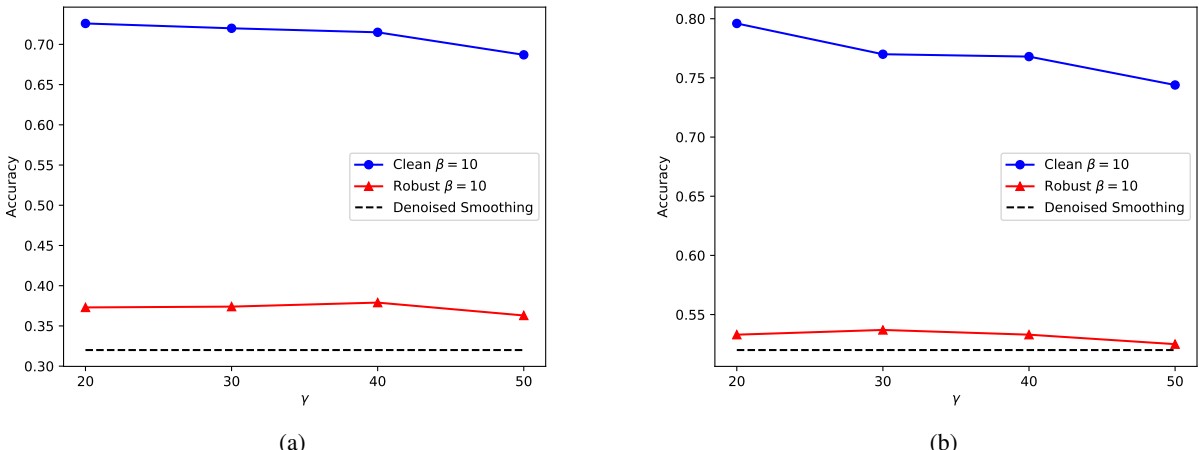

Figure S.5: Results for scale factor 8. Clean and robust accuracy for $\ell_\infty$ norm (a), and clean and robust accuracy for $\ell_2$ norm (b). We set $\alpha = 0.005$ for all the experiments. The dashed lines show the best accuracy for the denoised smoothing method (Salman et al., 2020). For the robust accuracy we only report 100 steps PGD attack accuracy.

| sigma | Accuracy under l-inf attack | Accuracy under l-2 attack | Clean accuracy |
|-------|------------------------------|----------------------------|----------------|
| 0.05  | 0                            | 0.1                        | 0.84           |
| 0.1   | 0.06                         | 0.36                       | 0.88           |
| 0.15  | 0.15                         | 0.47                       | 0.82           |
| 0.2   | 0.24                         | 0.47                       | 0.74           |
| 0.25  | 0.31                         | 0.51                       | 0.7            |
| 0.3   | 0.26                         | 0.46                       | 0.64           |
| 0.35  | 0.29                         | 0.41                       | 0.56           |
| 0.4   | 0.29                         | 0.41                       | 0.47           |
| 0.45  | 0.29                         | 0.39                       | 0.44           |
| 0.5   | 0.28                         | 0.42                       | 0.49           |

Table S.1: Empirical results of denoised smoothing (Salman et al., 2020) using the MSE objective for the denoiser. Pre-trained classifier is a Resnet-18, and denoiser is DnCNN.

| sigma | Accuracy under l-inf attack | Accuracy under l-2 attack | Clean accuracy |
|-------|------------------------------|----------------------------|----------------|
| 0.05  | 0                            | 0.1                        | 0.86           |
| 0.1   | 0.08                         | 0.38                       | 0.89           |
| 0.15  | 0.16                         | 0.49                       | 0.84           |
| 0.2   | 0.25                         | 0.5                        | 0.76           |
| 0.25  | 0.32                         | 0.53                       | 0.72           |
| 0.3   | 0.26                         | 0.48                       | 0.65           |
| 0.35  | 0.3                          | 0.42                       | 0.59           |
| 0.4   | 0.31                         | 0.4                        | 0.51           |
| 0.45  | 0.29                         | 0.39                       | 0.49           |
| 0.5   | 0.27                         | 0.39                       | 0.47           |

Table S.2: Empirical results of denoised smoothing (Salman et al., 2020) using the stability objective for the denoiser. Pre-trained classifier is a Resnet-18, and denoiser is DnCNN.

| number of ensemble models | Accuracy (%) | ResNet-18 (inc.) | ResNet-18 (exc.) | ResNet-34 | ResNet-50 | DenseNet-121 | VGG-16 |
|----|----|----|----|----|----|----|----|
| 1  | Clean  | 74.2 | 42.5 | 43.2 | 36.5 | 45.7 | 31.3 |
|    | Robust | 39.0 | 18.1 | 15.5 | 22.5 | 18.9 | 11.9 |
| 2  | Clean  | 88.5 | 88.2 | 88.5 | 88.6 | 88.8 | 89.6 |
|    | Robust | 27.7 | 27.3 | 24.4 | 23.0 | 28.5 | 29.1 |
| 4  | Clean  | 89.0 | 89.1 | 89.3 | 88.9 | 89.2 | 88.7 |
|    | Robust | 27.9 | 26.2 | 27.7 | 26.5 | 31.6 | 30.1 |
| 20 | Clean  | 90.1 | 89.8 | 89.6 | 89.1 | 89.6 | 88.9 |
|    | Robust | 28.2 | 26.9 | 28.0 | 26.9 | 31.9 | 30.8 |

Table S.3: Transferability of robustifiers trained with a single ResNet-18, and 2, 4, 20 Resnet-18 ensemble to another Resnet-18 with different parameters (Resnet-18 (exc.)), Resnet-34, Resnet-50, Densenet-121, and VGG-16 under $\ell_\infty$ attack. Accuracy of the robustifier with one of the Resnet-18 it trained on is shown in Resnet-18(inc.) column.

| number of ensemble models | Accuracy (%) | ResNet-18 (inc.) | ResNet-18 (exc.) | ResNet-34 | ResNet-50 | DenseNet-121 | VGG-16 |
|---|---|---|---|---|---|---|---|
| 1 | Clean | 82.2 | 42.5 | 43.2 | 36.5 | 45.7 | 31.3 |
| | Robust | 57.4 | 27.8 | 26.6 | 20.4 | 29.9 | 19.5 |
| 2 | Clean | 86.3 | 85.3 | 85.8 | 85.3 | 86.4 | 84.9 |
| | Robust | 45.4 | 43.5 | 44.3 | 42.2 | 47.9 | 47.3 |
| 4 | Clean | 88.4 | 88.1 | 88.4 | 88.3 | 88.4 | 87.8 |
| | Robust | 49.0 | 47.5 | 49.0 | 48.6 | 48.9 | 48.9 |
| 20 | Clean | 89.2 | 89.9 | 89.4 | 89.0 | 89.1 | 88.2 |
| | Robust 49.6 | 47.1 | 48.8 | 48.2 | 49.0 | 49.5 | 49.5 |

Table S.4: Transferability of robustifiers trained with a single ResNet-18, and 2 and 4 Resnet-18 ensemble to another Resnet-18 with different parameters (Resnet-18 (exc.)), Resnet-34, Resnet-50, Densenet-121, and VGG-16 under $\ell_2$ attack. Accuracy of the robustifier with one of the Resnet-18 it trained with is shown in Resnet-18(inc.) column.

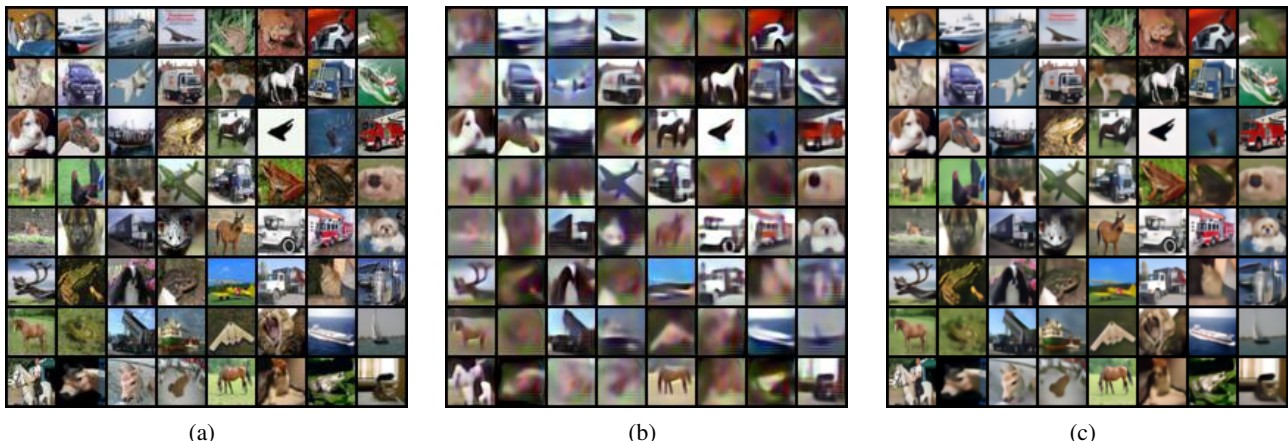

(a)                                    (b)                                    (c)

Figure S.6: Sample input (a) and output of robustifiers trained with different set of intermediate layers for computing activation in Eq. 5: first four layers (b) and first four layers with the last fully-connected layer (c). For both (b) and (c), loss coefficients are $\alpha = 0.005, \beta = 10, \gamma = 15$, threat model is $\ell_\infty$ attack. Clean and robust accuracy for (b) and (c) are 78% and 39%, and 87% and 31%, respectively.