# OpenReview forum: "Empirical robustification of pre-trained classifiers"
_ICML.cc/2021/Workshop/AML — ICML 2021 Workshop AML Poster_

### Official Review · Reviewer_jVQL · 2021-06-20

**Rating:** Accept
**Confidence:** 5

**Review:**

This is an interesting work that applies adversarial training to an input processing network to robustify pre-trained classifiers. Although the idea is simple, the paper shows its advantages over the denoising smoothing approach. Some losses are proposed to further improve the training of the robustifier. The proposed method is more effective than denoising smoothing, and also exhibits a good transferability between different models.

The idea of this work is very similar to a previous work [1], however, the citation and discussion are missing.

Besides, it's better to compare the performance with adversarial training.

[1] Liao et al., Defense against adversarial attacks using high-level representation guided denoiser. CVPR 2018.

---

### Decision · Program_Chairs · 2021-06-21

**Decision:**

Accept (Poster)

**Comment:**

This paper proposed an interesting method to robustify pre-trained classifiers. Despite the simplicity of the idea, the method is effective is many scenarios.